# Effectiveness of ongoing single dose rifampicin post-exposure prophylaxis (SDR-PEP) implementation under routine programme conditions—An observational study in Nepal

**Nand Lal Banstola**[1]*, **Epco Hasker**[2], **Liesbeth Mieras**[3], **Dambar Gurung**[4], **Bhuwan Baral**[5], **Suresh Mehata**[6], **Sagar Prasai**[7], **Yograj Ghimire**[8], **Brij Kumar Das**[9], **Prashnna Napit**[10], **Wim van Brakel**[11]

1 Programme department, NLR, Biratnagar, Koshi, Nepal, 2 Mycobacterial Diseases and Neglected Tropical Diseases, Institute of Tropical Medicine, Antwerp, Belgium, 3 Research and Advocacy, NLR IO, Amsterdam, The Netherlands, 4 Field programme, FAIRMED, Biratnagar, Koshi, Nepal, 5 Technical department, FAIRMED, Kathmandu, Bagmati, Nepal, 6 Public Health, Health office Morang, Biratnagar, Koshi, Nepal, 7 Public Health, Health office Sunsari, Ineruwa, Koshi, Nepal, 8 Public Health, Health office Jhapa, Bhadrapur, Koshi, Nepal, 9 Public Health, Health office Udayapur, Gaighat, Koshi, Nepal, 10 Leprosy Programme, Leprosy Control and Disability Management Section, Kathmandu, Bagmati, Nepal, 11 Medical Technical, NLR IO, Amsterdam, The Netherlands

* nbastola123@gmail.com, nandlalbanstola@nlrnepal.org.np

**Data Availability Statement:** All relevant data underlying the findings of this study are included

## Abstract

### Background/Introduction

Leprosy control remains a challenge in Nepal. Single-dose rifampicin post-exposure prophylaxis (SDR-PEP) shows promise in reducing leprosy incidence among contacts of index cases, contributing to reducing the transmission of *Mycobacterium (M.) leprae*. This study evaluates the effectiveness of routine SDR-PEP implementation in Nepal in addition to contact screening, focusing on its impact on reducing leprosy risk among contacts and potential population-level effects.

### Methodology

We conducted a retrospective cohort study to compare leprosy case notification rates and leprosy risk among close contacts. We compared two districts implementing SDR-PEP (the intervention group) and two without (the comparator group). Data from 2015 onwards included demographics, index case types, and contact relationships. Statistical analyses, including Cox regression and Kaplan-Meier survival curves, assessed the impact of SDR-PEP implementation.

### Findings

All four districts showed a decrease in case notification rates since 2015, with the steepest decline in the intervention districts. The risk of developing leprosy among contacts was

within the manuscript and its Supporting Information files. The data that support the findings of this study are publicly available from Mendeley with the identifier(s) DOI: 10.17632/kc4h4mh8n7.1 OR ACCESS ON https://data.mendeley.com/datasets/kc4h4mh8n7/1.

**Funding:** This research project is supported by the Leprosy Research Initiative (LRI) with NLB as PI, grant number FP_22\19/LRI. The funder had no role in the study design, data collection and analysis, publication decisions, or manuscript preparation.

**Competing interests:** The authors have declared that no competing interests exist.

significantly lower in the intervention districts (HR 0.28, 95% CI 0.18–0.44). SDR-PEP offered 72% protection, consistent over time, as shown in Kaplan-Meier plots. The protective effect was equally strong in blood-related contacts (HR 0.29 versus 0.27 in others, $p$ = 0.32), and the proportion of MB cases among incident cases was not significantly different post-PEP (51.4% vs. 53.6%, p = 0.82).

## Conclusions

This study demonstrates the substantial protective effect of integrating SDR-PEP in routine leprosy control programs with contact screening, significantly lowering leprosy risk among contacts. SDR-PEP is equally effective for blood-related contacts and does not preferentially prevent PB cases. While suggesting potential population-level impact, the study design does not allow for firm conclusions at this level. Further research is needed to fully assess SDR-PEP's effectiveness in diverse contexts and optimize its implementation. Integrating SDR-PEP within well-organized contact screening programs is effective and is expected to reduce leprosy transmission when applied as a rolling intervention.

### Author summary

Nepal continues to see new leprosy patients every year. It is known that close contacts of leprosy patients are at increased risk of developing leprosy and that providing them with a single dose of the antibiotic rifampicin as post-exposure prophylaxis (SDR-PEP) reduces that risk. This study looked at the impact of administrating SDR-PEP to screened contacts of leprosy patients as part of Nepal's routine leprosy control efforts. The intervention was implemented in two districts, and the results were compared to those of two districts where SDR-PEP was not provided. The findings showed a 72% reduction in leprosy risk among contacts in the districts using SDR-PEP. Importantly, SDR-PEP provided similar protection levels in blood-related as in non-blood-related contacts. These results underscore the role of SDR-PEP as a powerful component of routine leprosy control, contributing to stopping transmission. However, further studies are required to assess the population-wide impact and to optimise its implementation across diverse settings.

## Introduction

Leprosy, a chronic infectious disease caused by *Mycobacterium (M.) leprae*, remains a significant concern in regions worldwide, including Nepal [1,2]. For the past four decades, leprosy control has been based on early case detection and the provision of multi-drug therapy (MDT). However, in many countries, these interventions have not yet led to the interruption of transmission. Worldwide annual new case notifications have declined gradually and reached 174,087 in 2022. Despite substantial advances in leprosy control initiatives, persistent challenges remain in achieving early detection and preventing transmission. One new intervention in leprosy control is post-exposure prophylaxis (PEP) in the form of single-dose rifampicin (SDR). SDR-PEP is given to individuals in close contact with leprosy patients. Contacts of leprosy patients (household, social, neighbour) are known to be the group at the highest risk of developing leprosy [3]. Physical distance, genetic relationship, age, and leprosy classification are independent risk factors in contact with patients with leprosy [4]. Hence, leprosy control

strategies are now shifting their focus to prevention by targeting these high-risk groups with screening and administration of chemoprophylaxis [5,6].

The COLEP trial conducted in Bangladesh demonstrated a remarkable 57% reduction in leprosy incidence among contacts of newly diagnosed patients, documented two years after receiving SDR as prophylactic treatment [7]. The protective effect of SDR was maximal after the first two years, with no additional effect after four and six years. However, the total impact of the intervention was still statistically significant after six years ($p = 0.025$), and no rebound was observed in the SDR arm at a later stage.

The Leprosy Post-Exposure Prophylaxis (LPEP) Program was an international, multicentre feasibility study implemented within the leprosy control programs of Brazil, India, Indonesia, Myanmar, Nepal, Sri Lanka, Cambodia, and Tanzania [8]. LPEP explored the feasibility of combining three critical interventions: systematically tracing contacts of individuals newly diagnosed with leprosy, screening the traced contacts for leprosy, and administering SDR to eligible contacts. Outcomes were assessed regarding the number of contacts traced, screened, and provided SDR. The significant findings were that SDR-PEP is safe, can be integrated into different leprosy control programs with minimal additional effort once contact tracing has been established and is generally well accepted by index patients, their contacts, and healthcare workers [9]. Retrospective contact screening and SDR-PEP administration was also shown to be feasible in Cambodia, increasing the number of contacts receiving PEP at the start of the intervention [10]. These findings have been confirmed in other studies [11,12]. The LPEP Program has also invigorated local leprosy control programs through the availability of a prophylactic intervention; therefore, it was recommended to roll out SDR in all settings where contact tracing and screening have been established [9].

Different trials on PEP have yielded varying results. The pivotal COLEP trial in Bangladesh showed a 57% reduction in incidence over two years post-intervention without any rebound in the following years. A study in a high-incidence setting in Indonesia showed no effect of PEP provided to close contacts alone. Still, there was a clear effect of PEP when provided as a blanket measure to an entire island population [5]. High background incidence could have been the reason for the lack of effect of PEP provided to individual contacts. Recently, results of two new trials were published, one in a low endemicity setting in China using rifapentine or SDR, the other, the PEOPLE trial, in highly endemic villages in Comoros and Madagascar using a single administration of double the regular dose of rifampicin [12,13]. Both trials again showed a protective effect for SDR-PEP, with 41% protection (not significant; $p = 0.23$) in the China trial versus 45% protection ($p = 0.005$) in the PEOPLE trial [12,13].

During the LPEP Program in Nepal, a close contact approach was used, targeting mainly household contacts and neighbours, resulting in an average inclusion of 23 contacts per index case. The LPEP Program spanned four years (2015–2018). SDR-PEP was consistently provided in the selected areas and continued after the study period's end. Furthermore, a wealth of long-term data on close and distant contacts, such as household contacts, close relatives and neighbour contacts, and more distant associates, is available. Contact screening was also performed in districts not included in the LPEP Program, but no SDR-PEP was issued in these districts.

Despite adopting SDR-PEP as a national policy in Nepal, doubts about its effectiveness persist. These doubts hinder the widespread implementation of SDR-PEP. Within the Nepali context, these uncertainties primarily stem from a lack of evidence concerning the short- and long-term benefits of SDR-PEP within routine programme settings and its seemingly limited effectiveness among blood-related household contacts (24% protection) as demonstrated in the COLEP study [7].

The LPEP Program did not include follow-up of contacts and has therefore not provided additional evidence of the effectiveness of SDR-PEP to date. In particular, the effectiveness of implementing SDR-PEP as a 'rolling intervention', that is, continuously giving SDR to the contacts of every new case, has not been evaluated. The trials conducted to date have mostly given PEP once to a particular cohort of contacts. We expect that an ongoing PEP intervention will have increased effectiveness in reducing leprosy incidence compared to the once-only intervention approach.

Few studies have yet assessed the impact of incorporating SDR-PEP into a routine leprosy control programme or the impact of PEP on leprosy incidence at the population level [14]. Available evidence from research contexts shows that chemoprophylaxis with SDR-PEP effectively reduces the risk of developing leprosy among contacts [15]. Still, additional evidence is needed to demonstrate the impact of SDR-PEP when this strategy is implemented as part of routine leprosy control [7,14,16–18].

This retrospective cohort study aims to assess the effectiveness of ongoing (7–8 consecutive years) SDR-PEP implementation in districts in Nepal compared with districts without such intervention. The findings are expected to strengthen the evidence base for chemoprophylaxis with SDR-PEP and help develop strategies to incorporate PEP in routine programmes.

## Methods

### Ethical considerations

The Nepal Health Research Council (NHRC) reviewed and approved the research protocol and related documents. The Protocol Registration number is 238/2022 P, reference number 118, approved on 24 July 2022. Written formal consent was obtained from the participants involved. Parents/Guardians provided consent for children.

### Study design

We retrospectively compared cohorts of contacts of leprosy patients from the two districts that provided SDR-PEP (the intervention group) to contacts of all incident cases to cohorts of contacts from the two districts where no SDR-PEP was provided, the comparator group. The intervention started with the LPEP project, which was implemented from 2015 to 2018, and has since been continued. Our assessment was conducted from July 2022 to December 2023. These cohorts are from adjacent, comparable districts in the Koshi Province of Nepal (Fig 1).

Intervention and comparator districts were pre-selected and specified as such in the study protocol, but the intervention allocation was not randomised.

### Setting/location and respondent sampling and recruitment

Koshi province is situated in East Nepal. The intervention cohort was recruited among contacts who received SDR-PEP (150 mg, 300 mg, 450 mg or 600 mg as per age and weight) during the LPEP Program from 2015 to 2019 in the Jhapa and Morang districts, while the comparator cohort was recruited among contacts from Sunsari and Udayapur districts who did not receive any PEP. All four districts are part of Koshi province and are mostly flat, densely populated areas, possess a wide range of ethnic diversity, and are considered high-endemic for leprosy [19]. The population's socioeconomic status is predominantly low (poor and ultra-poor), with a moderate literacy rate [20]. Three districts, Parsa, Jhapa, and Morang, participated in the LPEP Program, which commenced in 2015. The SDR-PEP intervention was discontinued in Parsa in 2018, so we did not include this district in our analysis.

A mixed contact approach–household, neighbour, and social contacts of index patients– was used in the LPEP Program for the SDR-PEP implementation. The same contact types

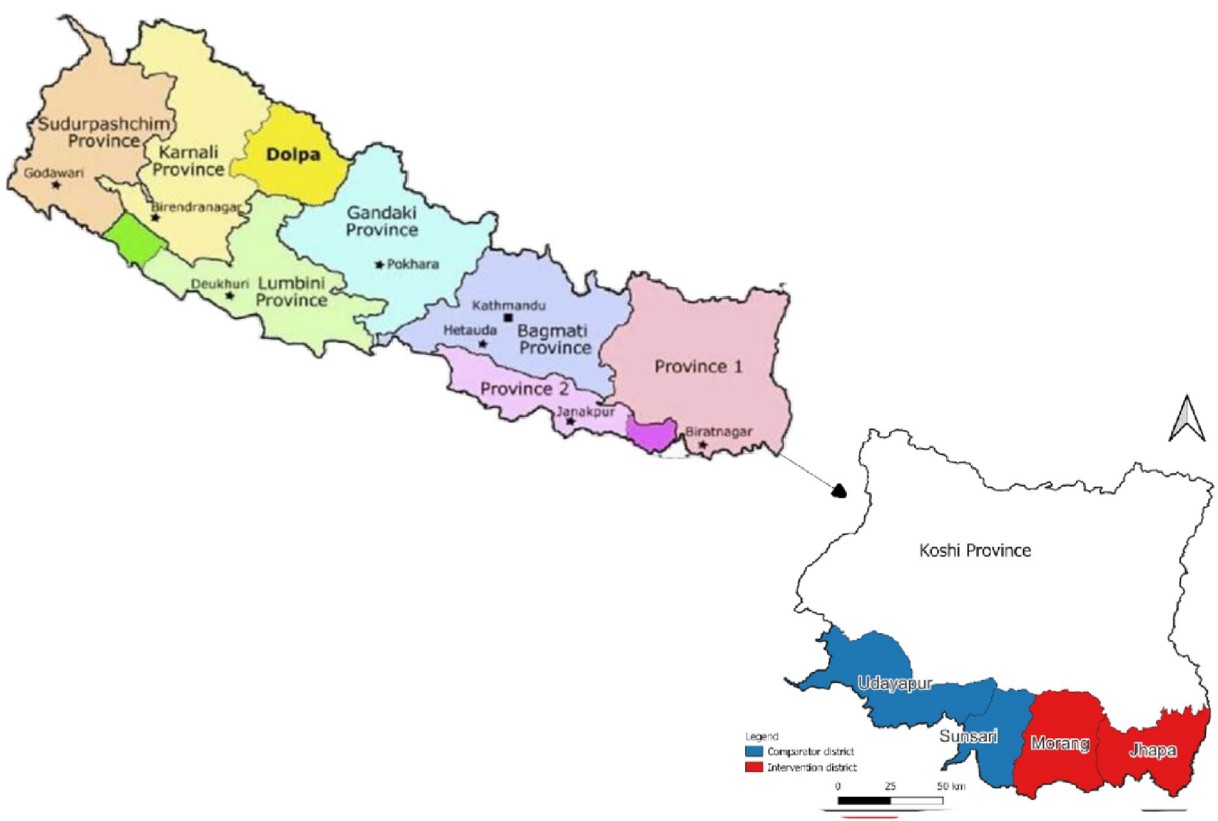

**Fig 1. The study districts (http://www.planiglobe.com/?lang=enl - PlaniGlobe, http://www.planiglobe.com, CC BY 2.0).**

were included in the comparator cohort of this study. Household contacts included all individuals living in the same household or compound as the index case for at least three months, using the same kitchen, and living under the same roof. Neighbour contacts included all individuals living in households/compounds adjacent to or otherwise close to the household/compound of the index case for at least 3 months. Social contacts were those who stayed together with an index case for at least 20 hours per week for at least 3 months.

The criteria for selecting the comparator districts were that they should be as comparable as possible to the intervention districts in all aspects other than PEP administration. For this reason, we selected the bordering districts Sunsari and Udayapur, which have similar socio-cultural, economic, educational, health, and disease-related conditions. In these districts, as in the LPEP intervention districts, a policy of active screening of close contacts of incident leprosy patients was implemented. Records were kept on all contacts screened. Population-wise, Udayapur is smaller than the other districts, with an estimated population in 2022 of 295,084 versus 892,139 for Jhapa, 1,165,892 for Morang, and 953,767 for Sunsari. In total, over the period of 2015–2019, 58,631 contacts of 2,516 index cases were screened across the four districts, 23 contacts per case on average. Table 1 provides a more detailed overview by district. All four districts' screenings were recorded in paper registers (S1 Annex), from which our study participants were selected.

## Sample size and sampling strategy

Based on the incidence rate among close contacts of leprosy patients observed in the control group of the COLEP trial in Bangladesh (3.35/1,000), we calculated the sample size required to

**Table 1. Overview of the study districts for the period 2015–2019.**

| District* | Mid period population | Leprosy cases 2015–2019 | Mean annual incidence rate per 1,000,000 | Contacts screened 2015–2019 | Screening ratio (contacts/index case) |
|---|---|---|---|---|---|
| Jhapa | 804,287 | 924 | 230 | 20,251 | 22 |
| Morang | 1,058,985 | 933 | 176 | 20,665 | 22 |
| Sunsari | 871,667 | 484 | 111 | 13,340 | 28 |
| Udayapur | 284,312 | 175 | 123 | 4,375 | 25 |

* Jhapa and Morang–Intervention; Sunsari and Udayapur–Comparator

Leprosy incidence has decreased in all four districts since 2015; the decline is most evident in Morang district, as shown in Fig 2. The incidence rate per one million in Morang decreased from 343 in 2015 to 127 in 2021, and in Jhapa, it decreased from 216 in 2015 to 76.2 in 2021.

Incidence among children (< 15 years) decreased in Morang and Sunsari and fluctuated in Jhapa and Udayapur (Fig 3).

detect a 50% rate difference. Using a power of 80% and a significance level of 5%, 14,100 contacts per study arm were required.

All 58,631 screened contacts in the four districts were entered from the contact registers in an Excel database. From this database, we drew a random sample of 45,306 from both study arms (Jhapa– 13,618, Morang– 13,681, Sunsari– 15,042 and Udayapur– 2,965). We then visited these contacts, going down the list from the beginning. If someone could not be contacted,

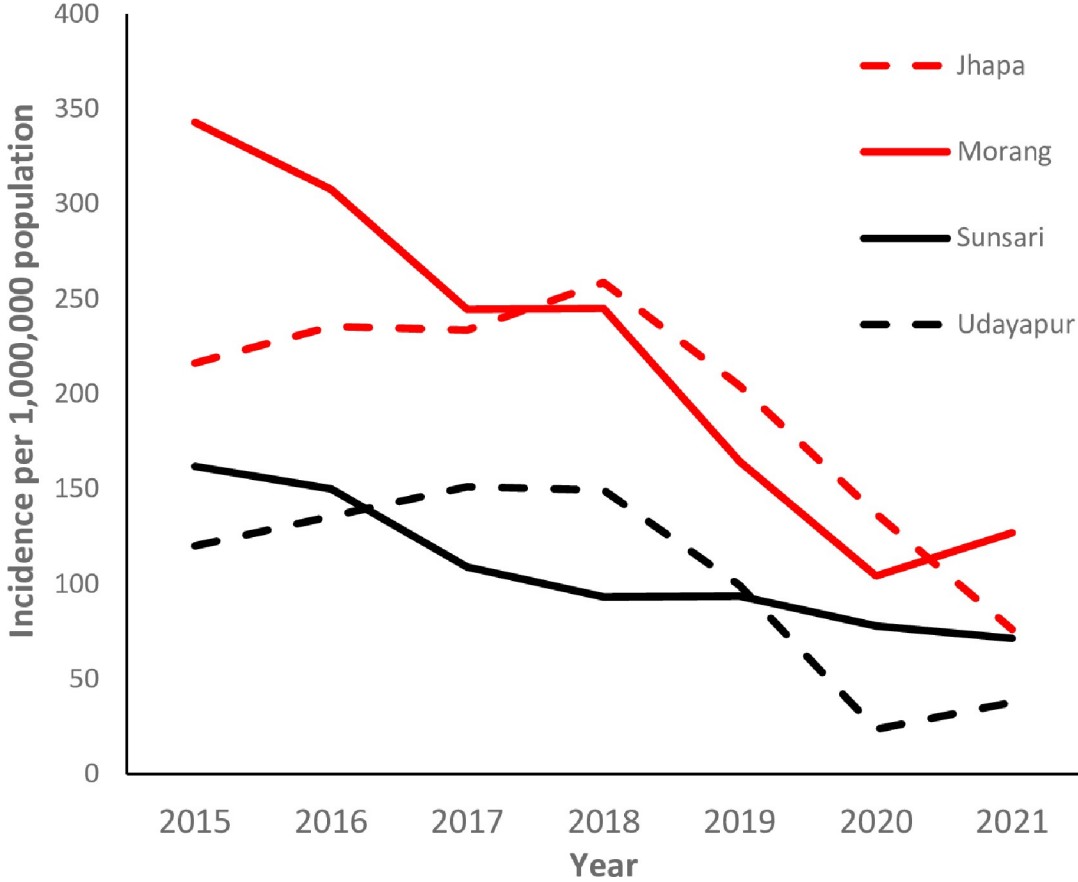

**Fig 2. Leprosy incidence in the four study districts (2015–2021).**

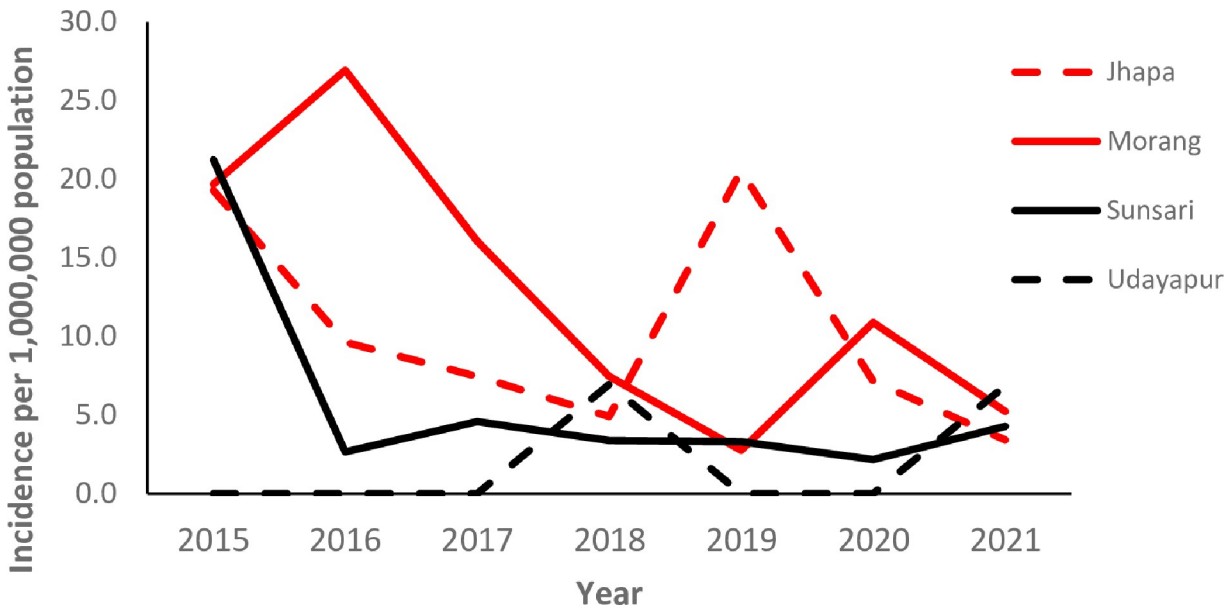

**Fig 3. Leprosy incidence among children in the four study districts (2015–2021).**

they would be excluded, and we moved to the next person on the list. We stopped recruitment when the target sample size of 14,100 in each arm had been reached.

## Outcome measures

The primary endpoint in this study was the number of contacts diagnosed with leprosy during follow-up in each of the study arms. The main outcome measure was the hazard rate ratio (HRR).

## Data collection

In all four districts, home visits were conducted, and contact screening was done under the direct supervision of a research coordinator. We defined and classified leprosy according to the National Leprosy Manual of Nepal. We strictly followed the three cardinal signs of leprosy defined by WHO and incorporated in the National Manual. For the quality assurance, we adopted a three-tier system of monitoring and confirmation:

- 'Leprosy detection teams' were established to visit homes and examine contacts for possible signs and symptoms of leprosy. These teams comprised community health volunteers and project staff (research assistants) working in pairs of male and female staff.

- A 'Leprosy confirmation team' was established in different clusters of municipalities to diagnose persons identified with possible signs or symptoms by the detection teams. This team comprised health workers, well-trained and experienced in leprosy. This team was equipped to take skin smears from all possible and confirmed cases and could consult a highly experienced dermatologist when needed. All team members received the same training, and their experience working in leprosy-endemic areas spanned at least eight years.

- A team of supervisors was established to monitor the diagnosed leprosy cases and link them to the nearest health centres for further care and treatment support.

Data were first collected on paper-based forms and checked for completeness and correctness. Then, using REDCap software, the data from these paper forms were transferred to an electronic database.

## Data management and analysis

From the REDCap database, data were extracted as CSV files and analysed in Stata /IC v18 for Windows. Demographic variables (gender, age, religion) and characteristics of index cases (MB/PB, blood-related) were analysed using proportions with percentages by district and by study arm. We also explored their effect as potential confounders. We calculated incidence rates by study district, excluding new cases diagnosed within three months of initial screening, assuming they already suffered from leprosy at the first screening. We fitted a Kaplan-Meier plot to compare the intervention to comparator districts. We then fitted Cox proportional hazards regression models with incident leprosy as the outcome variable, intervention as the main variable of interest, and each potential confounder as a secondary variable. As an exposure variable, we used total follow-up time per study arm. In this way, differences in follow-up times were adjusted for in the analysis. We used 'Tests of proportional-hazards assumption' provided by Stata to check whether the proportional hazards assumption was met. All potential confounders that changed the estimate of the primary exposure by more than 10% were tested in a multivariate model. We started from a saturated model and eliminated secondary exposures one at a time, starting from the weakest confounder included, until doing so would change the estimate of the primary exposure by more than 10%. As a sensitivity analysis, we repeated the Cox model without the comparator district with the highest incidence and the intervention district with the lowest incidence.

## Results

We enrolled a study sample of 28,453 contacts, 14,248 in intervention districts and 14,205 from comparator districts. The intervention districts were equally represented, with 7,072 participants from Jhapa and 7,176 from Morang. Among the comparator districts, Sunsari was overrepresented, with 12,151 participants, versus 2,054 from Udayapur. This, to some extent, compensated for the smaller population, number of leprosy patients and registered contacts in Udayapur, as shown in Table 1. In all districts, female participants outnumber male participants. The enrolled sample was slightly older in the intervention districts (Jhapa and Morang). This was related to the unavailability of child-friendly rifampicin formulations, resulting in a very low inclusion of the under-five age group in intervention districts. The proportion of MB index cases was higher in the comparator districts (Sunsari and Udayapur). Detailed information is provided in Table 2 below.

The intervention districts had longer average follow-up times: 84.1 and 80.3 months for Jhapa and Morang versus 64.4 and 64.2 months for Sunsari and Udayapur. In total, 123 incident leprosy cases were recorded. Compared to either of the intervention districts, the leprosy incidence rate among contacts enrolled was almost twice as high in Sunsari and almost nine times higher in Udayapur (Table 3).

We have excluded 17 incident cases from our analysis, all from comparator districts (8 in Sunsari and 9 in Udayapur), because they occurred within 3 months of the first screening. Of the 106 incident cases retained, 37 occurred in the intervention and 69 in the comparator districts. In the intervention districts, 19 (51.4%) were MB versus 37 (53.6%) in the comparator districts ($p$ = 0.82, Table 4).

We also excluded all contacts below five years of age at the first screening; there were no leprosy cases in this group. This resulted in an average annual incidence rate of 3.79 per 10,000

**Table 2. Characteristics of the study sample.**

| District<br>Factor | Jhapa | Morang | Sunsari | Udayapur |
|---|---|---|---|---|
| Sex | | | | |
| • Female | 3,862 (54.6) | 4,088 (57.0) | 6,815 (56.1) | 1,169 (56.9) |
| • Male | 3,210 (45.4) | 3,088 (43.0) | 5,336 (43.9) | 885 (43.1) |
| Age group | | | | |
| • 2–4 | 0 (0.0) | 2 (0.0) | 700 (5.8) | 86 (4.2) |
| • 5–9 | 195 (2.8) | 368 (5.1) | 1,219 (10.0) | 160 (7.8) |
| • 10–14 | 759 (10.7) | 855 (11.9) | 1,194 (9.8) | 167 (8.1) |
| • 15–24 | 1,545 (21.9) | 1,497 (20.9) | 2,356 (19.4) | 440 (21.4) |
| • 25+ | 4,573 (64.7) | 5,454 (62.1) | 6,682 (55.0) | 1,201 (58.5) |
| Religion | | | | |
| • Hindu | 6,192 (87.6) | 6,384 (89.0) | 10,529 (86.7) | 1,862 (90.7) |
| • Muslim | 286 (4.0) | 379 (5.3) | 1,331 (11.0) | 50 (2.4) |
| • Other* | 594 (8.4) | 413 (5.8) | 291 (2.4) | 142 (6.9) |
| Blood relationship | 2,006 (28.4) | 1,938 (27.0) | 2,060 (17.0) | 537 (26.1) |
| BCG scar | 6,083 (86.0) | 6,576 (91.6) | 11,046 (91.9) | 1,899 (92.5) |
| MB index case | 4,328 (61.2) | 4,191 (58.4) | 8,871 (73.0) | 1,520 (74.0) |

*Others–minority groups rather than Hindus and Muslims; **The numbers between brackets are %

***Jhapa and Morang–Intervention; Sunsari and Udayapur–Comparator

in the intervention districts versus 9.59 per 10,000 in the comparator districts. The Kaplan-Meier plot (Fig 4) shows a consistently higher hazard rate in the comparator districts ($p < 0.0001$). The test for the proportional hazards assumption resulted in a chi square of 0.10, $p = 0.75$, indicating that the assumption was met.

The crude hazard ratio for intervention vs comparator was 0.35 (95% CI 0.23–0.54). Blood-related contacts had a more than five times higher risk than other contacts ($p < 0.0001$), and contacts of MB index cases had an almost two times higher risk than contacts of PB index cases ($p = 0.0092$). Table 5 below provides more details. Another factor significantly associated with incident leprosy was religion, with those belonging to smaller religious minorities three to four times more at risk ($p < 0.0001$) than Hindus or Muslims.

In our multivariate model, we tested all factors that in a bivariate model with intervention as main exposure changed the hazard ratio of intervention by 10% or more. This was the case for religion and blood-related contact. Including both in the model changed the hazard ratio of the intervention to 0.25. The difference in the intervention effect estimate (without

**Table 3. Leprosy incidence among contacts and average follow-up time by district.**

| District | Contacts enrolled | Incident leprosy cases recorded | Average follow-up time (months) | Incidence rate per 10,000 per year (PYAR) |
|---|---|---|---|---|
| Jhapa | 7,072 | 18 | 84.1 | 3.63 |
| Morang | 7,176 | 19 | 80.3 | 3.96 |
| Sunsari | 12,151 | 48 | 64.4 | 7.36 |
| Udayapur | 2,054 | 38 | 64.2 | 34.60 |

*Note*: PYAR = Person-Years at Risk. Person-Years at Risk is calculated by multiplying the contacts by the average follow-up time (in years). The incidence rate per 10,000 PYAR provides a standardized measure of leprosy incidence, accounting for the number of contacts and the duration at risk. (*Jhapa and Morang–Intervention; Sunsari and Udayapur–Comparator*)

**Table 4. Incident leprosy cases by study arm and leprosy classification.**

| Intervention | Leprosy classification | | Total | P-value |
|---|---|---|---|---|
| | PB (%) | MB(%) | | |
| • Yes | 18 (48.6) | 19 (51.4) | 37 | 0.82 |
| • No | 32 (46.4) | 37 (53.6) | 69 | |
| Total | 50 (100) | 56 (100) | 106 | |

rounding) between the model with two additional terms (religion and blood-related contact) and the model with only blood-related contact was below 10%. We therefore retained only blood relationships as a confounder to keep the final model as simple as possible. The final estimate for the intervention thus became a hazard ratio of 0.28 (95% CI 0.18–0.44), i.e., a 72% reduction in risk during the follow-up period (Table 6).

We tested for interaction between intervention and being blood-related, but the interaction term was non-significant ($p$ = 0.32), and the protective effect of SDR-PEP among blood-related contacts (HR 0.28, 95% CI 0.17–0.48) was very close to the effect among non-blood-related contacts (HR = 0.27, 95% CI 0.12–0.57, Table 7).

As a sensitivity analysis, we excluded the comparator district with the highest incidence rate, Udayapur, and the intervention district with the lowest incidence rate, Jhapa. This resulted in a hazard ratio for the intervention of 0.39 (95% CI 0.22–0.71).

## Discussion

This observational study provides the first detailed assessment of an ongoing SDR-PEP intervention integrated into a routine leprosy control programme. At the individual level, we observed clear differences in the risk of developing leprosy among contacts between intervention and comparator districts. Contacts from intervention districts who had received SDR-PEP had a substantially lower risk, with an HR of 0.28 (95% CI: 0.18–0.44). In other words, the

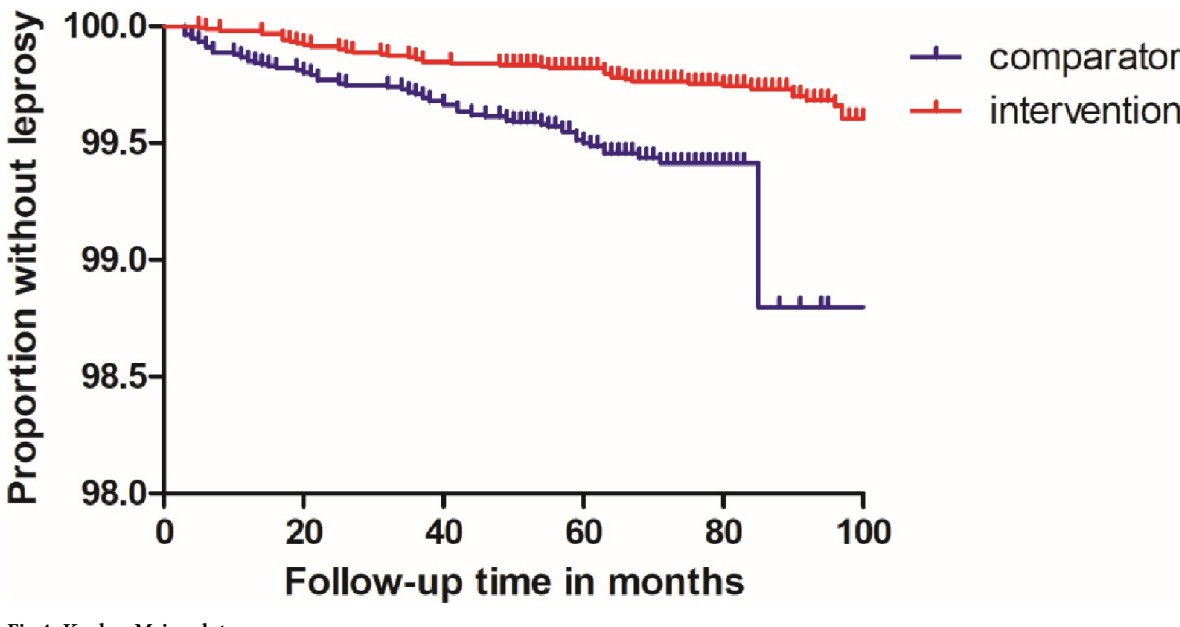

**Fig 4. Kaplan-Meier plot.**

**Table 5. Univariate associations with incident leprosy.**

| Factor | Total Enrolled | Incident cases (%) | HR (95% CI) |
|---|---|---|---|
| Intervention | | | |
| • Yes | 14,246 | 37 (0.26) | 0.35 (0.23–0.54) |
| • No | 13,402 | 69 (0.51) | Ref. |
| Sex | | | |
| • Female | 15,534 | 53 (0.34) | 0.78 (0.54–1.15) |
| • Male | 12,114 | 53 (0.44) | Ref. |
| Age group | | | |
| • 5–9 | 1,942 | 1 (0.05) | 0.14 (0.02–1.01) |
| • 10–14 | 2,973 | 10 (0.34) | 0.84 (0.43–1.61) |
| • 15–24 | 5,829 | 27 (0.46) | 1.16 (0.74–1.81) |
| • 25+ | 16,904 | 68 (0.40) | Ref. |
| Religion | | | |
| • Hindu | 24,312 | 84 (0.35) | Ref. |
| • Muslim | 1,912 | 5 (0.26) | 0.80 (0.33–1.98) |
| • Other | 1,424 | 17 (1.20) | 3.30 (1.96–5.57) |
| Blood-related | | | |
| • Yes | 6,381 | 64 (1.00) | 4.98 (3.37–7.35) |
| • No | 21,267 | 42 (0.20) | Ref. |
| BCG scar | | | |
| • Present | 24,804 | 91 (0.37) | 0.71 (0.41–1.22) |
| • Absent | 2,844 | 15 (0.53) | Ref. |
| Type index case | | | |
| • MB | 18,333 | 82 (0.45) | 1.83 (1.16–2.89) |
| • PB | 9,315 | 24 (0.26) | Ref. |

ongoing SDR-PEP intervention has shown a protective effect of 72% over an average follow-up time of 74 months. This finding underlines the effectiveness of the SDR-PEP intervention, along with active case detection, in reducing the risk of progression to clinical leprosy among (infected) contacts [21–27]. This effect was also observed at the population level in an earlier study from Morocco, revealing a statistically significant increase in the decline of annual case detection rates from 2012 onwards, when SDR-PEP was implemented as an ongoing intervention [28]. Our findings in a cohort of contacts in four Southeast Nepal districts align with these observations. Surveillance at the district level continues to assess whether, just as in Morocco, the SDR-PEP intervention contributes to reducing population-level leprosy incidence on top of the effect among contacts.

**Table 6. Multivariate associations with incident leprosy.**

| Factor | HR (95% CI) |
|---|---|
| Intervention | |
| • Yes | 0.28 (0.18–0.44) |
| • No | Ref. |
| Blood-related | |
| • Yes | 5.78 (3.91–8.56) |
| • No | Ref. |

**Table 7. Hazard ratio of intervention for blood-related contacts and others.**

| Factor | HR (95% CI) |
|---|---|
| Blood related | |
| • Yes | 0.29 (0.17–0.48) |
| • No | 0.27 (0.12–0.57) |

Several factors had a significant association with the observed reduction in risk of leprosy in the intervention cohort compared to the comparator cohort. Blood-related contacts were predominantly household members and exhibited a fivefold higher risk of developing leprosy [4]. Blood-related contacts were more common in the intervention arm, which led to some confounding that we corrected for in the final model. The risk of developing leprosy was higher if the index case was MB (crude HR 1.83, 95% CI 1.16–2.89). Although MB source cases were more common in the comparator districts, they were not a major source of confounding, so we omitted index case classification from the final model. Contacts from religious minority groups were at three to four times higher risk, which is likely to be related to their (lower) socio-economic status rather than religion per se (see Table 5).

Though our estimate of the protective effect of SDR-PEP was fairly conservative, excluding 17 early incident cases, all from comparator districts, it was very strong (72%). This is stronger than the effect observed in the COLEP trial (57%; 95% CI 32.9–71.9, $p$ = 0.0002) [7]. This is likely to be due to the ongoing nature of the PEP intervention in the current study, while SDR-PEP was only administered cross-sectionally in COLEP. Even when excluding the highest incidence comparator district and the lowest incidence intervention district, the protective effect of SDR-PEP was still strong (HR 0.39, 95% CI 0.22–0.71). The Kaplan-Meier plot shows a consistently lower risk of developing leprosy in SDR-PEP recipients over a period of more than six years. This represents the magnitude of the effect we can expect when contact screening combined with SDR-PEP provision is implemented under routine programme conditions.

Contact screening was already a component of the leprosy control programme in Nepal before the LPEP Program was implemented in 2015. The average number of contacts screened per index case was even higher in the comparator districts that had not been part of the LPEP study (Table 1). We observed that adding SDR-PEP involves minimal extra effort while providing substantial benefits in reducing the risk of disease progression and, thus, transmission [29,30].

Despite earlier suggestions that SDR-PEP might protect primarily against PB leprosy, in our study the proportion of incident MB cases was not significantly different among those who had received SDR-PEP than among those who had not (51.4% vs 53.6%, $p$ = 0.82) This confirms recent observations in the PEOPLE trial that PEP also protects against MB leprosy [11,30] and suggests that PEP protects against developing leprosy, irrespective of the type of disease the contacts might have been developing. Contrary to what was observed in the COLEP trial, the protective effect of PEP was comparable between blood-related contacts and other contacts (HR of 0.29 among blood-related contacts vs. 0.27 among non-blood-related contacts ($p$ = 0.32, Table 7) [7].

## Limitations

This study had limitations because it was a retrospective observational study, not a randomized controlled trial (RCT). The intervention districts were part of the Leprosy Post-Exposure Prophylaxis (LPEP) study. This may have resulted in more intensive case finding and fewer hidden cases, potentially impacting the incidence rates after the initial screening. In addition,

average follow-up time was shorter in comparator districts than in intervention districts, ~5½ vs. ~7 years, which may have resulted in more self-healing in the intervention districts [31]. However, the results are robust. We excluded 17 early cases from comparator districts. Even after excluding the best-performing intervention district and the worst-performing comparator district, the association between the intervention and protection against leprosy remained strong and statistically significant.

## Conclusion

This study underscores the strong protection offered to contacts of leprosy patients by adding SDR-PEP to contact screening in a routine leprosy control programme. The effect of SDR-PEP was equally strong in blood-related contacts, primarily household members, the prime target group for PEP interventions. Additionally, there was no difference in the proportions of MB and PB cases among incident leprosy cases who had and had not received PEP, indicating that the intervention did *not* selectively prevent PB leprosy.

The possible impact of SDR-PEP on transmission at population level is still being investigated. The design of this study does not permit definitive firm conclusions at that level. The individual protective effect of SDR-PEP is clear. Our findings suggest that if the intervention is sustained over several years, reducing individual risk of progression to disease among individual contacts, it will likely lead to a decrease in transmission as well.

## Supporting information

**S1 Annex. Recorded index cases and screened contacts.**
(DOCX)

## Acknowledgments

We thank all the PEP Impact Assessment Research Project staff, including Kinsha Gautam, Kritisha Kafle, Rubi Majhi, Binda Thapa, Ajit Kumar Mahato, and Prastab Paudel. Additionally, we thank the staff of NLR Nepal and FAIRMED, especially Madhav Bhatta, Labhi Shakya, Nirmala Sharma and Chiranjibi Nepal, for their dedicated efforts.

We also acknowledge the support provided by FAIRMED, READ Nepal, the Provincial Health Directorate Koshi, and the District Health Offices of Jhapa, Morang, Sunsari, and Udayapur. Finally, we recognize the valuable contributions and support from the health workers in the field and express our sincere gratitude to all the participants in this study.

## Author Contributions

**Conceptualization:** Nand Lal Banstola.

**Data curation:** Nand Lal Banstola.

**Formal analysis:** Nand Lal Banstola.

**Investigation:** Nand Lal Banstola.

**Methodology:** Nand Lal Banstola.

**Project administration:** Nand Lal Banstola.

**Resources:** Nand Lal Banstola.

**Software:** Nand Lal Banstola.

**Supervision:** Nand Lal Banstola.

**Validation:** Nand Lal Banstola.

**Visualization:** Nand Lal Banstola.

**Writing – original draft:** Nand Lal Banstola, Epco Hasker, Liesbeth Mieras, Dambar Gurung, Bhuwan Baral, Suresh Mehata, Sagar Prasai, Yograj Ghimire, Brij Kumar Das, Prashnna Napit, Wim van Brakel.

**Writing – review & editing:** Nand Lal Banstola, Epco Hasker, Liesbeth Mieras, Dambar Gurung, Bhuwan Baral, Suresh Mehata, Sagar Prasai, Yograj Ghimire, Brij Kumar Das, Prashnna Napit, Wim van Brakel.

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
