## [Decision Letter · Decision Letter 0]

16 Sep 2024

Dear Mr Banstola,

Thank you very much for submitting your manuscript "Effectiveness of ongoing single dose Rifampicin post-exposure prophylaxis (SDR PEP) implementation under routine programme conditions – an observational study in Nepal Impact assessment of single dose Rifampicin post-exposure prophylaxis" for consideration at PLOS Neglected Tropical Diseases. As with all papers reviewed by the journal, your manuscript was reviewed by members of the editorial board and by several independent reviewers. In light of the reviews (below this email), we would like to invite the resubmission of a significantly-revised version that takes into account the reviewers' comments. 

Dear Dr. Banstola and colleagues,

The reviewers and I found your manuscript to be important and worthwhile contribution on an important topic. There are specific requests to clarify the presentation made by Reviewer 2, in particular. Please address the ethical issues found in the reviews. Please add a study-specific protocol (referred to in the Methods, in addition to Ref 8) as a supplementary file and add a viable hyperlink, if available, to the protocol. As well as possible, please address the concerns in the detailed critique by Reviewer 3.

We cannot make any decision about publication until we have seen the revised manuscript and your response to the reviewers' comments. Your revised manuscript is also likely to be sent to reviewers for further evaluation.

Sincerely,

Paul J. Converse

Academic Editor

Elsio Wunder Jr

Section Editor

Dear Dr. Banstola and colleagues,

The reviewers and I found your manuscript to be important and worthwhile contribution on an important topic. There are specific requests to clarify the presentation made by Reviewer 2, in particular. Please address the ethical issues found in the reviews. Please add a study-specific protocol (referred to in the Methods, in addition to Ref 8) as a supplementary file and add a viable hyperlink, if available, to the protocol. As well as possible, please address the concerns in the detailed critique by Reviewer 3.

Reviewer's Responses to Questions

**Key Review Criteria Required for Acceptance?**

**Methods**

-Are the objectives of the study clearly articulated with a clear testable hypothesis stated?

-Is the study design appropriate to address the stated objectives?

-Is the population clearly described and appropriate for the hypothesis being tested?

-Is the sample size sufficient to ensure adequate power to address the hypothesis being tested?

-Were correct statistical analysis used to support conclusions?

-Are there concerns about ethical or regulatory requirements being met?

Reviewer #1: This is a very well-conducted study of great interest. The report is well-written.

Reviewer #2: - The sample size is sufficient to ensure adequate power to address authors' hypothesis

- However, some additional analysis could have been made:

o Page 8, Figure 1: is the decrease in leprosy incidence significant? “the decline is most evident in the Morang district” sentence should be replaced by a sentence given the incidence rates and the comparison between those from the different districts.

o Table 2: p-value should ne mentioned in order to easily identify the differences between the districts.

o Table 4: some p-value are 0.00 or 0.000, please clarify

- Page 6: 

o the dosage of RIF should be mentioned, 

o the possibility that people from one district travel in another should be discussed as this might interfere with the way to contract leprosy

- Page 7: could the authors comment on the contact average par case that seems very high (i.e. 23) in comparison to literature and also specify the nature of their relationship, as it was shown previously that single-dose rifampin offered significant protection for only neighbors of neighbors plus social contacts (reference 7).

- Page 10: regarding the diagnosis methods: the authors should clearly specify that the diagnosis was only based on clinic and provide some evidence that the abilities of the leprosy confirmation teams working in each district to diagnose leprosy were similar.

Reviewer #3: (No Response)

**Results**

-Does the analysis presented match the analysis plan?

-Are the results clearly and completely presented?

-Are the figures (Tables, Images) of sufficient quality for clarity?

Reviewer #1: The results are clearly presented.

Reviewer #2: - Tables and figures: 

o it should be clearly indicated which districts represent the comparator and the intervention groups in all tables and figures

o there are many small tables, it should be possible to merge some of them (e.g. Tables 2 and 3, Tables 4 and 5)

- Page 6: a map of Nepal highlighting the districts would be highly appreciated

- “discussion”: 

o page 16: the first sentences of the discussion should be replaced by more quantitative statements, the same can be said for the sentence “The individual protective effect of SDR-PEP is clear” page 19.

o given the results section, is not straightforward to see where the “protective effect of 72%” come from

o page 18: the authors should also comment on the risk of drug resistance selection in leprosy by given a single drug

Reviewer #3: (No Response)

**Conclusions**

-Are the conclusions supported by the data presented?

-Are the limitations of analysis clearly described?

-Do the authors discuss how these data can be helpful to advance our understanding of the topic under study?

-Is public health relevance addressed?

Reviewer #1: The Conclusions are fullt supported by the data presented.

Reviewer #2: Please refer to the above sections

Reviewer #3: (No Response)

**Editorial and Data Presentation Modifications?**

Reviewer #1: The Introduction is a very clear summary of current knowledge and practice in regard to post-exposure prophylaxis (PEP) for leprosy. I suggest adding a sentence about retrospective case-finding and PEP, which was done in Cambodia, with good results. This is probably a useful way to enhance the effect of PEP at the start of an intensive PEP programme. See: "Preventing leprosy with retrospective active case finding combined with single-dose rifampicin for contacts in a low endemic setting: results of the Leprosy Post-Exposure Prophylaxis program in Cambodia. Cavaliero, et al, Acta Tropica, 2021.

A second point relates to cases in children, which are regarded as a reasonable proxy for the level of transmission in a community. In Figure 1 of the current paper, the overall incidence rate of leprosy in each of the four districts is presented. If available, I would like to see the same Figure replicated to show the incidence in children under 15 years of age. Since there were only 123 incident cases in the study, the numbers may be too low to reach any conclusion, but it is an indicator that should be watched closely.

Reviewer #2: Page 4: the authors should provide a table summarizing the design, number of cases and main findings of the PEP trials already published together with their one study

Reviewer #3: (No Response)

**Summary and General Comments**

Reviewer #1: High quality study and paper.

Reviewer #2: This study represents an enormous piece of work given the challenges of such clinical trials, especially among population whose socioeconomic status is predominantly low (poor and ultra-poor) and when the follow-up needs to be long as for leprosy. Therefore, I believe that after the modifications requested this paper deserves to be published, as it will add valuable data regarding PEP implementation in the leprosy field.

Reviewer #3: (No Response)

PLOS authors have the option to publish the peer review history of their article (what does this mean?). If published, this will include your full peer review and any attached files.

Reviewer #1: Yes: Paul Saunderson

Reviewer #2: No

Reviewer #3: No
---

## [Editor Report · Decision Letter 1]

9 Nov 2024

Dear Mr Banstola,

We are pleased to inform you that your manuscript 'Effectiveness of ongoing single dose Rifampicin post-exposure prophylaxis (SDR PEP) implementation under routine programme conditions – an observational study in Nepal Impact assessment of single dose Rifampicin post-exposure prophylaxis' has been provisionally accepted for publication in PLOS Neglected Tropical Diseases.

Best regards,

Paul J. Converse

Academic Editor

Elsio Wunder Jr

Section Editor

Shaden Kamhawi

co-Editor-in-Chief

Paul Brindley

co-Editor-in-Chief

Dear Dr. Nand Lai Banstola,

Thank you for your careful and painstaking responses to the reviewers comments. Congratulations on the acceptance of your paper for publication in PLoS Neglected Tropical Diseases.

---

## [Editor Report · Acceptance letter]

27 Nov 2024

Dear Mr Banstola,

We are delighted to inform you that your manuscript, "Effectiveness of ongoing single dose rifampicin post-exposure prophylaxis (SDR-PEP) implementation under routine programme conditions – an observational study in Nepal," has been formally accepted for publication in PLOS Neglected Tropical Diseases.

Best regards,

Shaden Kamhawi

co-Editor-in-Chief

Paul Brindley

co-Editor-in-Chief
